# Potential of Documentary Evidence to Study Fatalities of Hydrological and Meteorological Events in the Czech Republic

**Rudolf Brázdil** [1,2,*], **Kateřina Chromá** [2], **Jan Řehoř** [1], **Pavel Zahradníček** [2,3], **Lukáš Dolák** [1,2], **Ladislava Řezníčková** [1,2] and **Petr Dobrovolný** [1,2]

1.  Institute of Geography, Masaryk University, 611 37 Brno, Czech Republic; 433735@mail.muni.cz (J.Ř.); dolak@sci.muni.cz (L.D.); ladkar@sci.muni.cz (L.Ř.); dobro@sci.muni.cz (P.D.)
2.  Global Change Research Institute, Czech Academy of Sciences, 603 00 Brno, Czech Republic; kchroma@email.cz (K.C.); zahradnicek.p@czechglobe.cz (P.Z.)
3.  Czech Hydrometeorological Institute, 616 67 Brno, Czech Republic
*   Correspondence: brazdil@sci.muni.cz

**Abstract:** This paper presents the potential of documentary evidence for enhancing the study of fatalities taking place in the course of hydrological and meteorological events (HMEs). Chronicles, "books of memory", weather diaries, newspapers (media), parliamentary proposals, epigraphic evidence, systematic meteorological/hydrological observations, and professional papers provide a broad base for gathering such information in the Czech Republic, especially since 1901. The spatiotemporal variability of 269 fatalities in the Czech Republic arising out of 103 HMEs (flood, flash flood, windstorm, convective storm, lightning, frost, snow/glaze-ice calamity, heat, and other events) in the 1981–2018 period is presented, with particular attention to closer characterisation of fatalities (gender, age, cause of death, place, type of death, and behaviour). Examples of three outstanding events with the highest numbers of fatalities (severe frosts in the extremely cold winter of 1928/1929, a flash flood on 9 June 1970, and a rain flood in July 1997) are described in detail. Discussion of results includes the problem of data uncertainty, factors influencing the numbers of fatalities, and the broader context. Since floods are responsible for the highest proportion of HME-related deaths, places with fatalities are located mainly around rivers and drowning appears as the main cause of death. In the further classification of fatalities, males and adults clearly prevail, while indirect victims and hazardous behaviour are strongly represented.

**Keywords:** fatality; fatality features; documentary data; hydrological and meteorological event; Czech Republic

## 1. Introduction

Recent climate change, largely represented by global warming caused by increasing concentrations of greenhouse gases due to human activities, is accompanied not only by changes in the mean values of climatic variables, but also by changes in the frequency and severity of hydrological and meteorological events (HMEs). Projections of HMEs for the future based on climate models are particular points of attention (e.g., [1–4]). Furthermore, observational evidence of HMEs, including their impacts, has great importance in approaches to risk management and cannot be ignored. This was emphasised by the Sendai Framework for Disaster Risk Reduction 2015–2030 (SFDRR), adopted at the Third United Nations (UN) World Conference on Disaster Risk Reduction in Sendai (Japan) on 18 March 2015, outlining targets and priorities "to prevent new and reduce existing disaster risks" [5].

HMEs cause great material damage and take many human lives every year worldwide, as documented by reinsurance agencies' statistics of great natural disasters; one example is Munich Reinsurance Company [6]. Although the number of deaths in Europe, with 8.9% of all fatalities in 1980–2017, remains far behind Asia (71.1%) and is somewhat less than North America (13.7%), it still represents a very serious problem. Heat waves are of particular concern; in Europe, c. 70,000 fatalities were attributed to heat waves during July–August 2003 (especially in France, Germany, Italy, Portugal, Romania, Spain, and the United Kingdom (UK); e.g., [7]), while some 56,000 fatalities were reported for July–September 2010 in Russia (especially in the Moscow region, Kolomna, Mokhovoye, Voronezh, Ramonskiy, Maslovka) [8]. However, other individual HMEs, including floods and windstorms, for which the number of victims is only tens or hundreds on a European scale (e.g., [9,10]), may be very significant at national or regional levels.

A high degree of attention to loss of human lives arising out of HMEs with respect to particular features is devoted to flood fatalities as, for example, in Australia (e.g., [11–13]), the United States (e.g., [14–17]), India (e.g., [18]) and China (e.g., [19]). In Europe, flood fatalities are combined with those associated with landslides (e.g., [20] or [21] on the global scale), and the majority of papers focus on the Mediterranean region (e.g., [22–26]). Especially worthy of note among the European studies is a paper by Petrucci et al. [27], which was presented to the Mediterranean Flood Fatalities (MEFF) database, which contains detailed data concerning flood fatalities in five Mediterranean regions for the 1980–2015 period. More recently, this database was extended to nine regions for the 1980–2018 period, as the European Flood Fatalities (EUFF) database [28].

In addition to flood fatalities, those due to landslides, lightning, windstorms, hail, or storm surges in the Calabria Region (southern Italy) were analysed for the 2000–2014 period by Aceto et al. [29] and for the 2000–2016 period by Petrucci et al. [30]. Badoux et al. [31] investigated fatalities related to several natural hazards in Switzerland for the 1946–2015 period. Lightning fatalities and injuries going back to 1852 in the United Kingdom were studied by Elsom [32] and for 1988–2012 by Elsom and Webb [33] (see also [34] for a broader, worldwide overview). Recently, Heiser et al. [35] reported the Austrian torrential event catalogue. Since 2016, the European Severe Storms Laboratory publishes annual reports, including the number of fatalities associated with severe weather in particular European countries. Based on these reports, 217 fatalities were reported in 2016, 294 in 2017, and 356 in 2018 [36–38].

In the Czech Republic, several important floods and flash floods, responsible not only for extensive material damage but also for many fatalities, occurred since the second half of the 1990s: July 1997 [39], July 1998 [40], August 2002 [41], March–April 2006 [42], June–July 2009 [43], May and August 2010 [44], and June 2013 [45]. However, these contributions generally cited only total mortality and lacked any more detailed information. Matters were similar in the reporting of large windstorms, such as "Kyrill" in January 2007, "Emma" in March 2008 [46], and "Herwart" in October 2017 [47]. Only Brázdová [48] dealt in greater detail with selected fatalities, in an attempt to develop a simple model for estimation of fatalities during floods in the Czech Republic.

Although fatalities associated with individual more disastrous, or recent, events may be quite well documented at the state and regional administration level, publicly available and long-term systematic evidence is generally absent. This is especially valid for such data for the more distant past, for which information must be garnered from documentary evidence. The nature of such evidence may vary from information concerning HMEs, used in historical climatology [49,50] and historical hydrology [51,52], which report casualties and fatalities, as well as related material damage. These constitute highly important sources of knowledge with great potential for further use in the Czech Republic, as well as in many other European countries.

This study aims to demonstrate the potential of past and recent documentary data for the study of fatalities arising out of various HMEs. Preliminary data for the 1981–2018 period in the Czech Republic are analysed to demonstrate the spatiotemporal distribution and, in part, the demographic structure of fatalities, further complemented by detailed examples of HMEs involving outstanding numbers of fatalities during the 20th century. This is the first study to address HME-related fatalities in the Czech Republic in context and detail; it also contributes to the few existing European studies (beyond the Mediterranean) that investigate fatalities arising from the various types of HME. Section 2 describes the basic types of documentary data, with examples, and covers the methodology of their analysis. The spatiotemporal variability of fatalities over the territory of the Czech Republic with respect to different types of HMEs and relevant features of such fatalities, including analysis of three the most disastrous events of the 20th century, are presented in Section 3. Section 4 discusses uncertainties in documentary data, factors influencing fatalities, and results obtained in the broader context. The last section presents some concluding remarks.

## 2. Materials and Methods

### 2.1. Societal and Climatic Background

Because the 1981–2018 period in a particular country is used as an example of the potential of documentary evidence in studying fatalities associated with HMEs, some basic information about the Czech Republic should be included as background. The Czech Republic is located in central Europe. It has a territorial area of 78,870 km$^2$. The number of inhabitants in mid-2018 was 10.626 million, a figure that corresponds to an overall density of 135 inhabitants per km$^2$. Corresponding values in mid-1981 were slightly lower: 10.303 million inhabitants and 131 inhabitants per km$^2$. The relative number of people living in large cities (populations of over 100,000) is c. 22%, and the number of inhabitants of small municipalities (populations under 10,000) is 53%. Altogether, there are over 6200 municipalities in the Czech Republic. The age structure breaks down into 16% children (below 15 years) and almost 20% seniors (above 65 years). The female/male proportion slightly favours the female (50.8% to 49.2%, respectively), but this is changing slightly with time (e.g., in 1993 the figures were 51.4% and 48.6%, respectively) [53].

In terms of climatic patterns in 1981–2018, mean annual temperature was 8.1 °C (from 1.7 °C to 10.9 °C at individual meteorological stations) and mean annual precipitation totalled 692 mm (from 442 mm to 1467 mm). Annual variation of temperature exhibits a simple wave with a minimum in January (−1.8 °C) and a maximum in July (18.0 °C) (Figure 1a). During the period studied, temperatures showed a statistically significant increase (at the 0.05 significance level), with a linear trend of 0.44 °C per decade (Figure 1b). The annual distribution of precipitation is more complex, with a minimum in February (37 mm) and a maximum in July (89 mm); several other secondary minima and maxima appear (Figure 1a). Despite relatively high inter-annual variability, precipitation totals indicate relatively stable patterns with a non-significant linear trend (Figure 1c). One remarkable feature is the alternation of years with notable floods (1997, 2002, 2006, 2009, 2010, and 2013) and droughts (2000, 2003, 2007, 2012, 2015, and 2018).

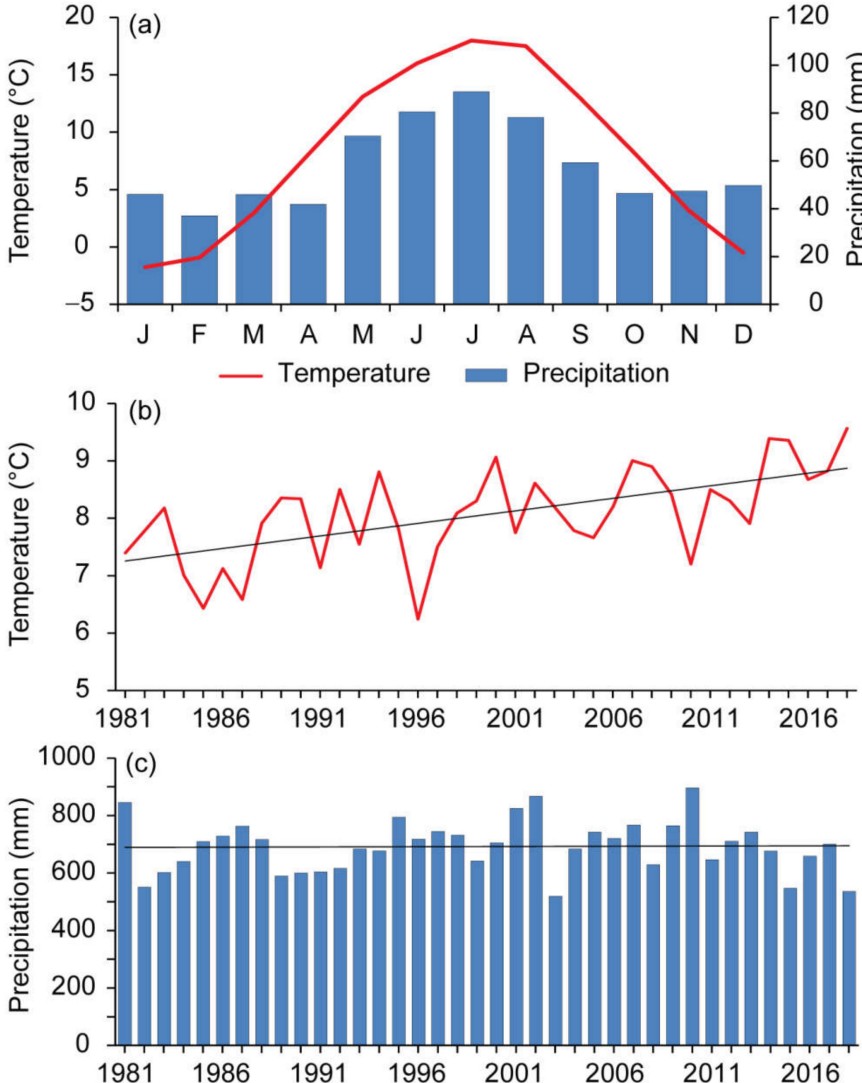

**Figure 1.** Climatic patterns of the Czech Republic in the 1981–2018 period, based on mean Czech temperature and precipitation series: (**a**) annual variations in mean monthly temperatures and precipitation; (**b**) fluctuations in mean annual temperature series; (**c**) fluctuations in annual precipitation series (series in (**b**) and (**c**) are supplemented by linear trends).

*2.2. Documentary Data*

Systematic collection of information concerning casualties suffered during HMEs over the territory of the recent Czech Republic started in 1901. It may be obtained from various types of documentary evidence, the basic types of which, together with examples of fatality reports, appear below.

2.2.1. Narrative Sources (Chronicles, "Books of Memory")

Local chronicles were kept in a range of settlements for the past in the Czech Lands, extensively extended into the 20th century, particularly after the establishment of independent Czechoslovakia in 1918 and legal enactment of Decree No. 80/1920 Coll. on municipal "books of memory" [54]. They contain broadly approached records related to life and events in the given place, including HMEs and their impacts. For example, the book of memory for the municipality of Uherčice describes two fatalities during a flash flood (for the location of places included herein, see Figure 2) on 4 May 1929 [55]: "*The cloud came from Palava* [the Pavlovské vrchy Hill] *and it was not so large; but in half an hour the lower part of the village was* [inundated by] *water,* [in the form of] *downburst, rain, and hail. Water ran to a depth of up to a metre and everything that stood in its way was swept off.* [ . . . ] *In house No. 69, the abode of Jakub*

*Švarc, a wagon stood in a shed. The gate to the street was closed and the courtyard was full of water. Jakub Švarc, who wanted to allow the water to flow off more rapidly, climbed onto the wagon in order to open the gate. But at that moment the water smashed the gate and swept away the wagon together with Švarc; once the water had gone down, he was found drowned. [ . . . ] František Furch, from house No. 42, was on his way from ploughing on 'Staré hory', together with his wife and their three-year-old son. Water [overflowing] into a basin from the village swept away his wagon, cows drowned, and Furch and his wife were carried off by the water. [ . . . ] [Also] a boy was taken by the water and on the subsequent day people found him drowned in sludge in the barn of Jan Smékal at No. 80.*"

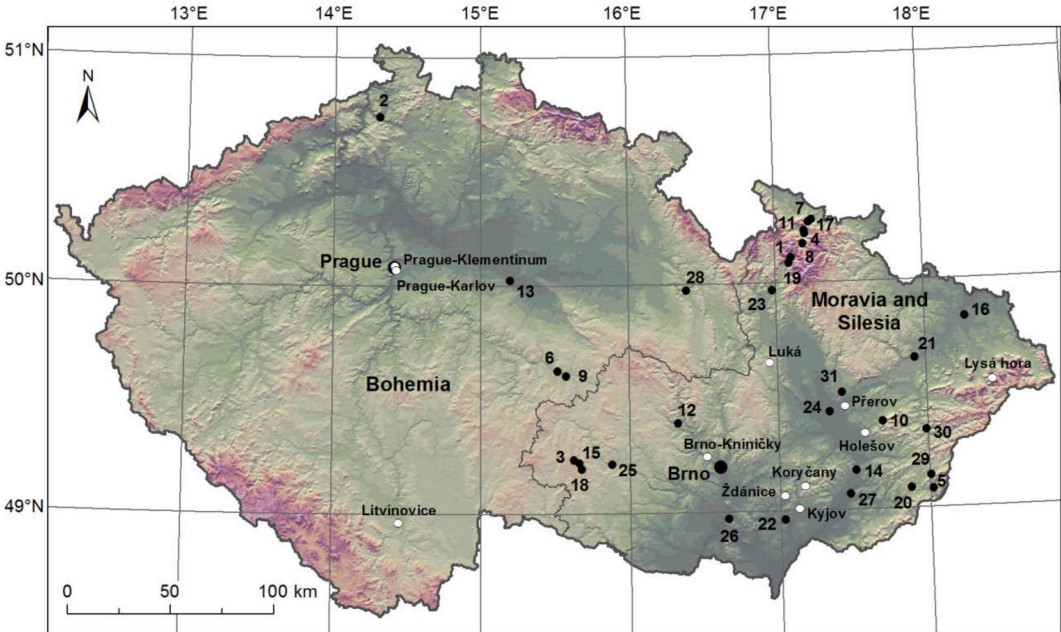

**Figure 2.** Meteorological stations (circles) and locations (points with numerals) in the Czech Republic mentioned in this paper (given in an alphabetical order): 1—Annín (today Loučná nad Desnou), 2—Benešov nad Ploučnicí, 3—Brtnička, 4—Bukovice (today Jeseník), 5—Bylnice (today Brumov-Bylnice), 6—Chlístov (today Okrouhlice), 7—Česká Ves, 8—Domašov (today Bělá pod Pradědem), 9—Havlíčkův Brod, 10—Hostýn, 11—Jeseník, 12—Jilmoví (today Pernštejnské Jestřabí), 13—Kolín, 14—Napajedla, 15—Opatov, 16—Ostrava, 17—Písečná, 18—Předín, 19—Rejhotice (today Loučná nad Desnou), 20—Slavičín, 21—Suchdol nad Odrou, 22—Šardice, 23—Šumperk, 24—Troubky, 25—Třebíč, 26—Uherčice, 27—Uherské Hradiště, 28—Ústí nad Orlicí, 29—Valašské Klobouky, 30—Vsetín, 31—Zábeštní Lhota.

### 2.2.2. Weather Diaries

In addition to records kept by the existing network of meteorological stations, certain people made individual observations of the weather and included them in private weather diaries. The records of Josef Lukotka in Vsetín during the 1903–1923 period may serve as an example [56]. For instance, his notes for 23 June 1908 include the following [57]: "*Lightning set a barn on fire at Bylnice; lightning in* [Valašské] *Klobouky killed a mother with grandson and lightning set a house on fire in the Slavičín region. To date our district has been spared such accidents. On this day* [22 June] [ . . . ] *28 people were killed by lightning in Moravia.*"

### 2.2.3. Newspapers and Other Media

Newspapers, conveying topical information of particular political, socio-economic, and societal interest, also report the occurrence of significant HMEs, often publishing information covering damage and fatalities. For example, two fatalities were reported in *Právo* newspaper [58] in association with a windstorm on 28 March 1997: "*Windstorms, which on Friday* [28 March] *swept over the* [Czech] *republic,*

*gusted at up to 180 km per hour, tore away roofs and uprooted trees. Under them* [the trees] *two people, 23-year-old driver D. S. from Prague in the Kolín region and a 70-year-old man in the Havlíčkův Brod region, came to untimely ends.* [ . . . ] *The* [young] *man was driving in a Mercedes car with his father and they were forced to stop by a fallen tree in the road.* [ . . . ] *They tried to remove this obstacle, but meanwhile, with an ear-splitting roar, a second tree fell. The young driver perished in situ below it.* [ . . . ] *The* [70-year-old] *man took to the forest* [near the village of Chlístov] *just at the moment when the wind raged strong enough to fell a tree and its descending crown unhappily struck the victim on the head and he died* [ . . . ]." In recent decades, newspaper reports may be supplemented by information taken from other media, such as the television, the internet, social networks, and others.

### 2.2.4. Parliamentary Proposals

Certain deputies of the Czechoslovak parliament proposed sets of measures to help people in the areas affected by particular natural disasters. These proposals are archived in the digital parliament library [59]. Such proposals contain basic information about some events and may also report fatalities, as an example from 11 May 1927 [60] demonstrates: "*On 9 May 1927 an extraordinary cloudburst occurred in the Třebíč region, devastating the entire cadastres of the villages of Předín, Opatov, Brtnička, and others. Floods of water did great damage to fields, roads, and bridges. A torrent of water a metre deep swept away the family of the cooper Štancl when they were returning from the fields; Štancl's wife Antonie and their two small children, aged three and a half and five and a half, were drowned. Their corpses were washed away by the rapid current and found only on the morning of the subsequent day* [10 May]." However, in the majority of other deputies' proposals, any fatalities are reported in rather general terms, and information about them has to be sourced elsewhere.

### 2.2.5. Epigraphic Evidence

Epigraphic evidence, consisting most frequently of water marks (e.g., [61,62]), may also contain information related to fatalities. For example, a memorial to the victims of the July 1997 flood in Troubky contains details of all nine fatalities (Figure 3a). Another example may be found on a crest of a road near the village of Jilmoví, commemorating the death of a young man by lightning strike and featuring a short inscription (Figure 3b): "*Stanislav Synek, at the age of 21 years, died by lightning strike at this spot on 16 July 1927*".

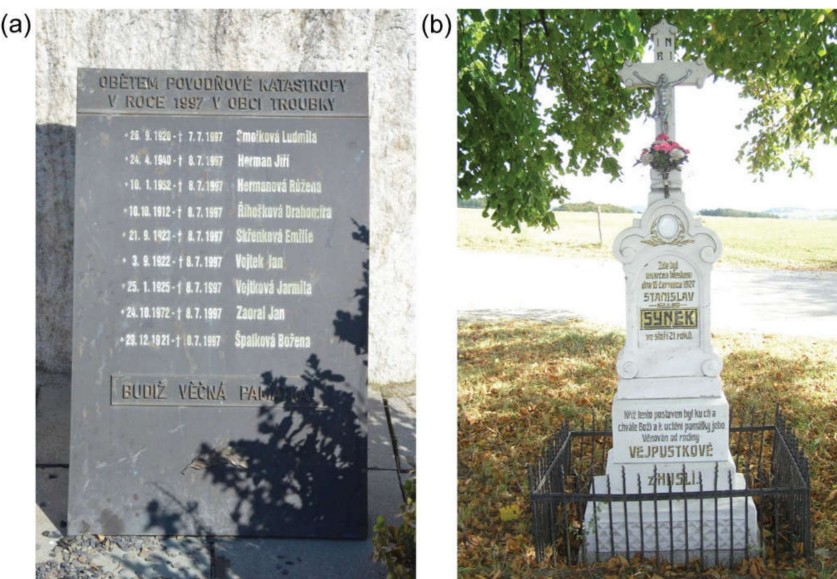

**Figure 3.** Examples of epigraphic evidence: (**a**) a list of victims of the July 1997 flood on the memorial at Troubky [63]; (**b**) report on the death of Stanislav Synek by lightning strike on a crest in the road near Jilmoví [64].

2.2.6. Systematic Meteorological and Hydrological Observations

Although the national networks of meteorological and hydrological stations naturally occupy themselves primarily with obtaining quantitative data about meteorological and hydrological elements and phenomena, their observers sometimes add short descriptions of damage and fatalities during particular events, often accompanied by related articles in local newspapers. For example, an observer at the Přerov climatological station recorded on 12 August 1933 that lightning during afternoon thunderstorms "*killed a woman in Zábeštní Lhota*" [65]. According to the February 1956 record-sheet of the Holešov climatological station [66], "*Several persons in Holešov and its surroundings (Svatý Hostýn, Napajedla, Uherské Hradiště) died during frosts around 10 February.*"

2.2.7. Professional Papers

Some descriptions of fatalities may be found in a few professional papers, describing individual disastrous events. For example, Polách and Gába [67], analysing the history of floods in the Šumperk and Jeseník Districts, described a disastrous flash flood on 1 June 1921: "*Cloudburst in the evening on 1 June* [1921]. *Huge landslides of 'mury'* [debris flow] *occurred. The Hučivá Desná valley became dammed and created a large temporary lake; the obstruction subsequently burst and did catastrophic damage throughout the river valley, right up to Šumperk. Four fatalities: At Annín two workers—Josef Langer (17 years) and Raimund Korger (36 years)—drowned during rescue work. In Rejhotice, the 6-year-old daughter of Mrs. Gabrielová, as well as schoolgirl Elsa Gabrielová, perished. Destructive consequences of the flood were also recorded around the River Bělá. The termination of seven human lives was the worst penance: two persons perished in Domašov, three in Bukovice (one of the victims was Emmi Plischkeová, a scholar at business school in the Monastery of Ursulines in Jeseník because her parents had a house in Bukovice close to the River Bělá), and one each in Česká Ves and in Písečná.*"

*2.3. Meteorological Data*

In addition to documentary data concerning HMEs fatalities, the following meteorological data from a number of stations of the Czech Hydrometeorological Institute (CHMI) are used herein:

(i)     Daily minimum temperatures $T_{min}$ from the Prague-Klementinum station (191 m above sea level (asl)) for 1 December 1928–15 March 1929 and mean $T_{min}$ in the 1961–1990 period;

(ii)    Daily snow depths at the Prague-Karlov (260 m asl) and Luká (510 m asl) stations for 1 December 1928–15 March 1929;

(iii)   Daily precipitation totals for selected meteorological stations in southeastern Moravia for 9 June 1970;

(iv)    Precipitation totals for 5–8 July 1997 for selected meteorological stations in the eastern part of the Czech Republic;

(v)     Monthly and annual temperature and precipitation series averaged over the territory of the Czech Republic for the 1981–2018 period [68].

*2.4. Methods*

Data concerning fatalities in which various HMEs were responsible for direct or indirect deaths were collected from documentary sources. Newspaper reports and internet data were taken as primary sources, supplemented by the records kept by climatological stations and some secondary sources (books, professional reports, or publications). Such items of information were firstly critically evaluated with respect to the reliability of the given source (preference given to primary sources). The quality of the report was also considered, with priority was given to detailed description(s) of the event and the circumstances that led to the corresponding death(s), as well as to information derived from personal losses; these were all preferred to only brief or generally summarising reports. Fatality data that appeared unreliable in the light of the above were eliminated from this study. As follows from Table 1, nearly three-quarters of all fatalities (74%) were documented from only one report and 16% from

two reports. Taking into account only individual types of documentary sources, nearly two-thirds of fatalities were extracted exclusively from (printed) newspapers (65.8%), followed by combined sources (15.2%) and the internet (11.9%). Whenever a fatality was covered by more than one report, the corresponding information was cross-checked; however, in the overwhelming majority of such cases, no direct contradictions emerged and differences were only negligible in largely irrelevant points of detail (e.g., with respect to a victim's age).

**Table 1.** Numbers of fatalities related to individual hydrological and meteorological events (HMEs) in the Czech Republic during the 1981–2018 period, categorised by numbers of fatality reports and the type of documentary sources from which fatalities were extracted.

| Type of HME | Number of Fatalities | | | | | | | | |
|---|---|---|---|---|---|---|---|---|---|
| | Total | According to Reports | | | | According to Sources | | | |
| | | 1 | 2 | 3 | More | Newspaper | Internet | Other | Combined |
| Flood | 91 | 66 | 17 | 8 | – | 63 | – | 13 | 15 |
| Flash flood | 54 | 25 | 17 | 1 | 11 | 17 | 10 | 6 | 21 |
| Windstorm | 31 | 24 | 1 | 2 | 4 | 17 | 9 | – | 5 |
| Convective storm | 22 | 18 | 4 | – | – | 16 | 6 | – | – |
| Lightning | 8 | 7 | 1 | – | – | 5 | 3 | – | – |
| Frost | 16 | 16 | – | – | – | 16 | – | – | – |
| Snow/glaze ice | 37 | 33 | 4 | – | – | 35 | 2 | – | – |
| Heat | 2 | 2 | – | – | – | 2 | – | – | – |
| Other HMEs | 8 | 8 | – | – | – | 6 | 2 | – | – |
| Total | 269 | 199 | 44 | 11 | 15 | 177 | 32 | 19 | 41 |
| % Total | 100 | 74.0 | 16.4 | 4.1 | 5.6 | 65.8 | 11.9 | 7.1 | 15.2 |

For further statistical analysis, critically evaluated fatality data were included in a newly created Czech fatality database, largely motivated by the concept of the MEFF database published by Petrucci et al. [27]. For each fatality, our database includes date, locality, type of HME, part (hour) of the day, name of the victim, their gender (male, female), age (exact in years or estimated: child, teenager, adult, senior), cause of death (e.g., drowning, falling tree or branches) and immediate environment (e.g., field, forest, home, street), type of fatality (direct, indirect), behaviour (normal and hazardous), and source of information. For example, on the evening of 4 July 2009 at c. 6:00 p.m., in the municipality of Benešov nad Ploučnicí, 40-year-old Naděžda Rubnerová drowned in the River Ploučnice in a flash flood. She was carried away by a torrent of water (direct victim) while trying to help save a wheelchair-bound woman (non-hazardous behaviour) [69]. Because such a comprehensive set of information is not necessarily available for every fatality, some parts of related information are categorised as absent. Collating fatalities for the individual HME events, the number of people injured, as well as fatalities/injured in the other countries, was also added to the database.

Fatalities were attributed to nine groups of HMEs, as shown below. In many cases they do not represent hydrological or meteorological extremes in the statistical sense (e.g., they are not based on return periods or low percentiles of corresponding distribution), but only those HMEs during which a fatality, or fatalities, occurred.

(i)　Flood: This includes rainfall-induced floods originating from single-day or multiple-day rainfall caused by precipitation-rich synoptic situations. Snow-melt floods arising after sudden melting of deep snow cover in winter/spring and mixed floods caused by a combination of snowmelt and rainfall (may be accompanied by ice jams on rivers) in winter or early spring are part of this group.

(ii)　Flash flood: Flash floods appear after the occurrence of torrential rains or cloudbursts (of high intensity and relatively short duration), usually accompanying thunderstorms.

(iii)　Windstorm: Strong winds related to large horizontal gradients of air pressure, of duration from a few hours to some days.

(iv)　Convective storm: Strong winds of short duration related to the development of cumulonimbus cloud (e.g., squall, tornado, downburst) with significant local/sub-regional effects.

(v)　Lightning: Lightning, as a phenomenon accompanying thunderstorms, may set fire to structures and trees, or kill or injure people by a direct strike.

(vi)　Frost: Very severe frosts when their effects may be accompanied by snowstorm or snowdrifts.

(vii)　Snow/glaze-ice: This includes, in particular, cases of snow calamity and the creation of glaze-ice on communications leading to car accidents.

(viii)　Heat: Extremely high temperatures, usually during heat waves.

(ix)　Other extremes: HMEs with fatalities that cannot be unequivocally attributed to any of the previous types (e.g., avalanche, thunderstorms in general).

## 3. Results

### 3.1. Spatiotemporal Variability of Fatalities

A total of 269 fatalities in the courses of 103 HMEs were documented in the Czech Republic for the most recent 1981–2018 period. This breaks down into averages of 2.6 fatalities per HME, 7.1 fatalities per year, and 2.7 HMEs per year. The highest number of fatalities (91) was recorded for floods, responsible for 33.8% of all fatalities (Figure 4). Because these were followed by 54 fatalities attributable to flash floods (20.1%), these two types of HME were associated with more than half of all fatalities (145 fatalities, i.e., 53.9%). The other important group of fatalities was related to strong winds (53, i.e., 19.7%), represented partly by windstorms (31, i.e., 11.5%) and partly by convective storms (22, i.e., 8.2%). Snow/glaze-ice calamities led to the deaths of 37 persons (13.8%). Further HMEs, such as lightning, heavy frost, or heat, together with HMEs not included in any of the above eight groups, led to the remaining 34 fatalities (12.6%).

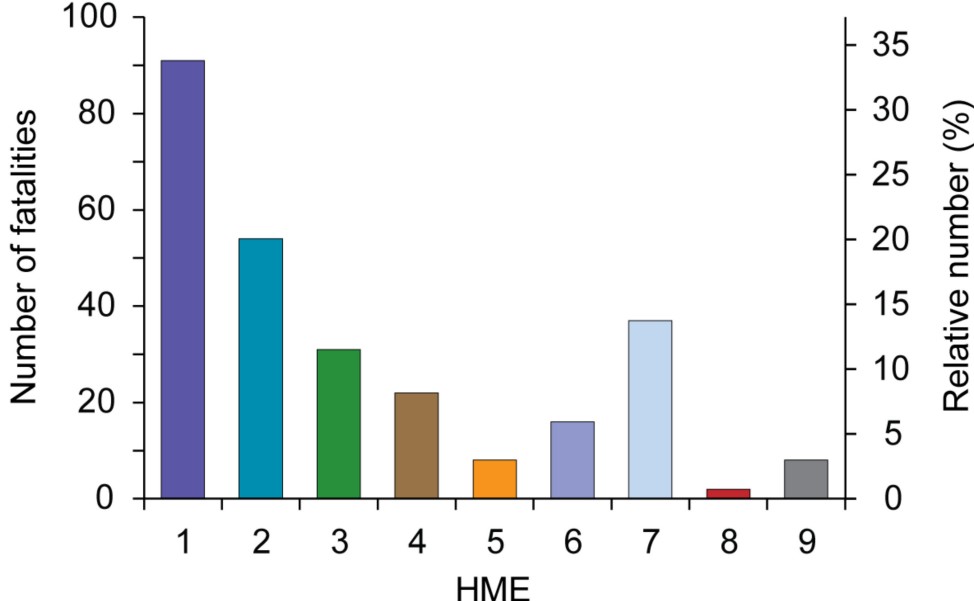

**Figure 4.** Absolute and relative numbers of fatalities related to individual types of hydrological and meteorological event (HME) in the Czech Republic during the 1981–2018 period: 1—flood, 2—flash flood, 3—windstorm, 4—convective storm, 5—lightning, 6—frost, 7—snow/glaze-ice, 8—heat, 9—other HMEs.

Because no fatalities were recorded in 10 years studied (Figure 5), the mean annual rate of fatalities in only years with fatal HMEs (3.7 HMEs per year) increased to 9.6 persons. Years with no fatalities occurred in the 1980s (three years), in the early 1990s (four years), and in the 2010s (three years), while

in another 10 years only one or two fatalities were recorded. The year 1997 was the worst of the group, with 44 documented fatalities, particularly associated with the flood of July 1997. Other years with higher numbers of flood fatalities lay far behind this catastrophic year (e.g., 27 fatalities in 2002, 19 in 2009, 18 in 2010, and finally 17 in 2013). A total of 23 fatalities was also recorded in 2007.

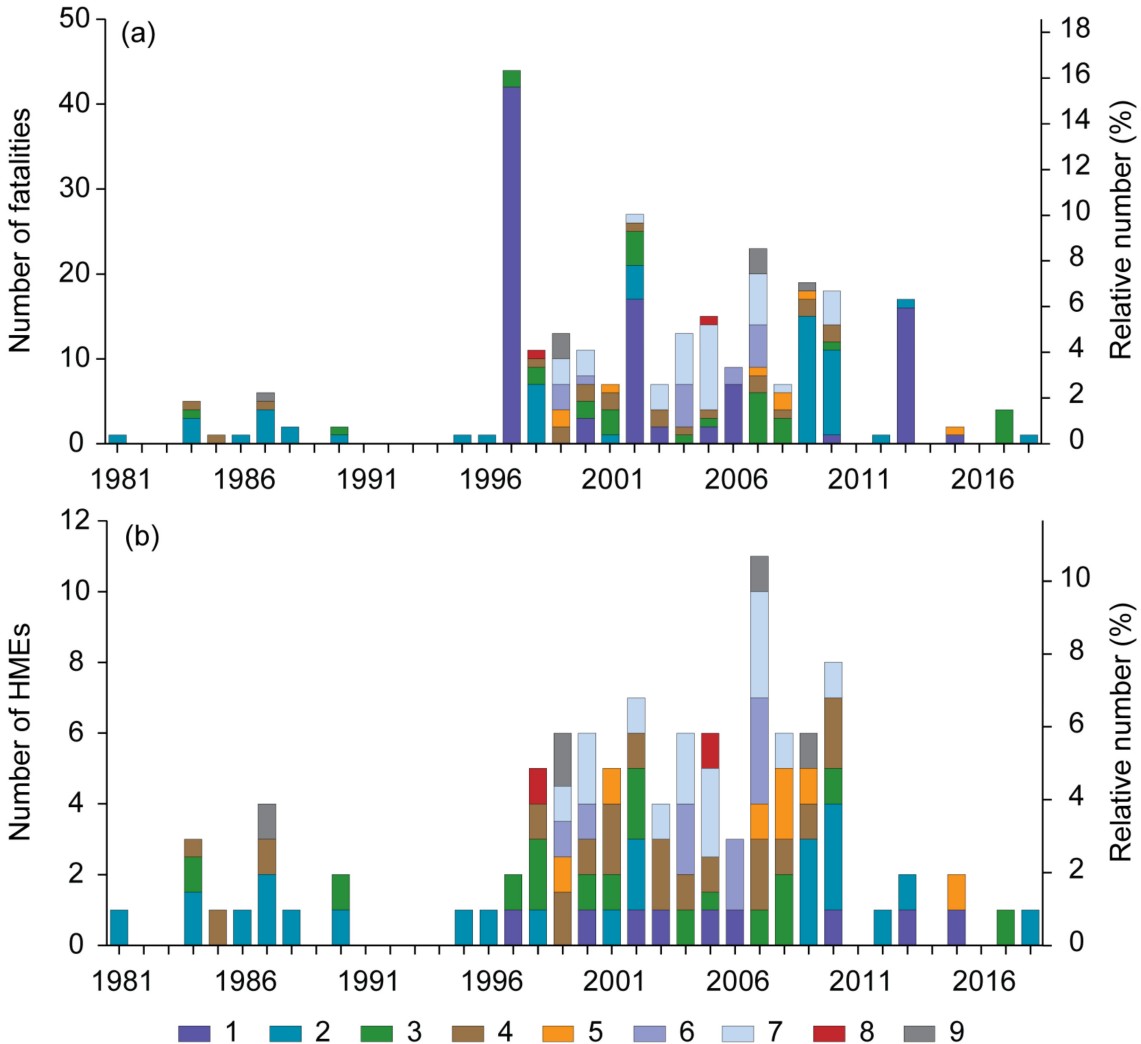

**Figure 5.** Fluctuations (**a**) in the annual numbers of fatalities and (**b**) in the annual numbers of HMEs with fatalities in the Czech Republic during the 1981–2018 period: 1—flood, 2—flash flood, 3—windstorm, 4—convective storm, 5—lightning, 6—frost, 7—snow/glaze-ice, 8—heat, 9—other HMEs.

The annual variation of 269 documented fatalities (Figure 6) showed a clear maximum in July (75 fatalities, i.e., 27.9% of the total number), followed by June (46, i.e., 17.1%), January (41, i.e., 15.2%), and August (32, i.e., 11.9%). The highest values of fatalities in the summer (153, i.e., 56.9%) were clearly attributable to events during rain floods and flash floods in 1997, 2002, 2009, 2010, and 2013, accounting for a total of 105 fatalities (i.e., 39.0%). The secondary maximum in January related especially to accidents during snow/glaze-ice calamities (22 fatalities, i.e., 8.2%), or people frozen to death during episodes of heavy frost. The latter two types of HME aligned with the winter months to total 63 fatalities (i.e., 23.4%) before spring (37, i.e., 13.8%) and autumn (16, i.e., 5.9%).

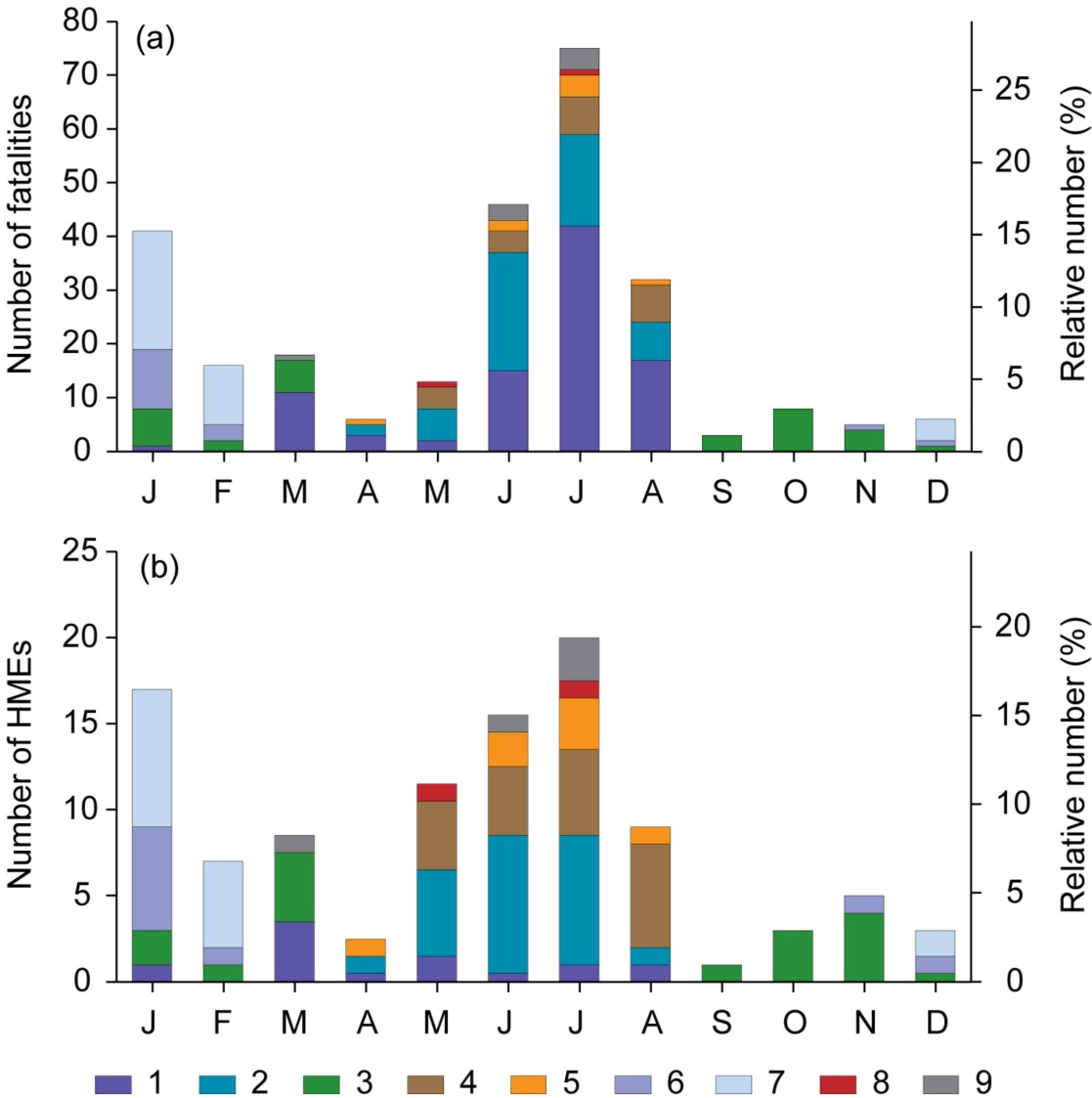

**Figure 6.** Annual variations (**a**) in monthly numbers of fatalities and (**b**) in the monthly numbers of HMEs with fatalities in the Czech Republic during the 1981–2018 period: 1—flood, 2—flash flood, 3—windstorm, 4—convective storm, 5—lightning, 6—frost, 7—snow/glaze-ice, 8—heat, 9—other HMEs.

The spatial distribution of fatalities over the territory of the Czech Republic (Figure 7) concentrated around rivers and watercourses, which followed from the greatest proportion of fatalities during floods and flash floods in the period analysed (see Figure 4). The locations of other places with fatalities were somewhat random. One fatality was documented for 185 municipalities, two fatalities for 28 municipalities, three for five municipalities, and four for three municipalities. The highest numbers of fatalities were found for Ostrava (six), Prague (eight, but with one occurring in the Prague-west district), and Troubky (nine).

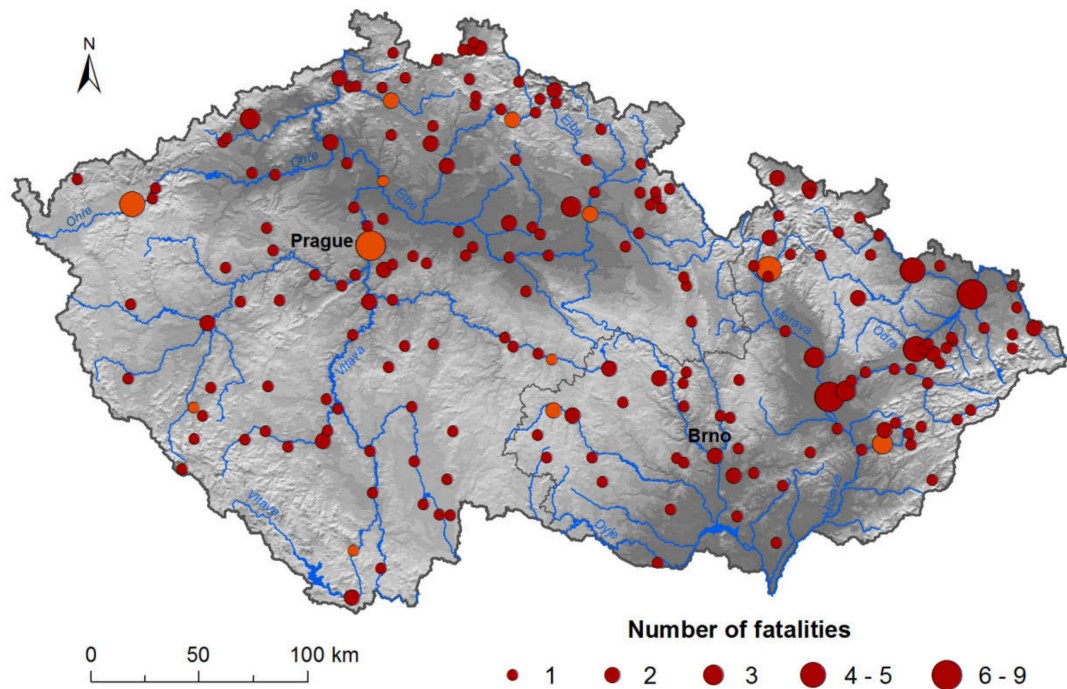

**Figure 7.** Spatial distribution of settlements with some fatalities during HMEs over the territory of the Czech Republic in the 1981–2018 period (settlements that served as centres of report for a whole district appear in light red).

*3.2. Features of Fatalities*

Since further data about fatalities during individual HMEs were available, it became possible to add more detailed information related to specific features of the fatalities (Figure 8). However, in a number of cases, the information acquired was of such a general nature, or even absent, that they were then classified as "unknown". The following six groups of fatality feature may be analysed:

3.2.1. Gender (Figure 8a)

A total of 181 male fatalities made up 67.3% of all documented deaths. The share of female fatalities was far smaller (64, i.e., 23.8%). Gender was not specified in sources for a further 24 fatalities (8.9%). The prevalence of male fatalities appeared clearly in all HME types except convective storms.

3.2.2. Age (Figure 8b)

Although a relatively high proportion of fatalities were among people of unknown age (85, i.e., 31.6%), elsewhere, the adult category (16–65 years) clearly prevailed with 131 fatalities (i.e., 48.7%). The number of seniors (>65 years) was much higher than the number of children (0–15 years): 44 (16.4%) and nine (3.3%), respectively. Such a distribution also remained in place for all individual HME types. Applying more detailed age categories [25], 18.2% of all fatalities fell into a category of 30–49 years, followed by 65–84 years (16.4%), 50–64 years (15.2%), and 15–29 years (14.1%). The proportion of children (0–14 years) in this study was higher (3.3%) than fatalities at the age of 85+ years (1.1%).

3.2.3. Cause of death (Figure 8c)

Drowning related to floods and flash floods clearly dominated cause of death with 111 fatalities (i.e., 41.3%), ahead of those associated with car accidents (42, i.e., 15.6%) in snow/glaze-ice calamities, with falling trees or branches (41, i.e., 15.2%) during windstorms and convective storms far behind. No other reason—collapse of a building, freezing to death, lightning strike, unknown—rose above 6%. Particular health issues (heart failures, hypothermia in cold water) were responsible for 22 fatalities (8.2%).

### 3.2.4. Place of Death (Figure 8d)

For more than one-fifth of fatalities (60, i.e., 22.3%), the place of death was unknown. Rivers and their banks (66, i.e., 24.5%) were relatively easily classified. Also, roads, with death occurring in cars, were important (53, i.e., 19.7%). Deaths within buildings and those taking place in built-up areas (streets, spaces outside buildings, gardens, etc.) accounted for 30 fatalities each (i.e., 11.2%). The landscape category (fields, meadows, forests, campsites, allotments or recreational cottage areas, etc.) saw a total of 24 fatalities (8.9%).

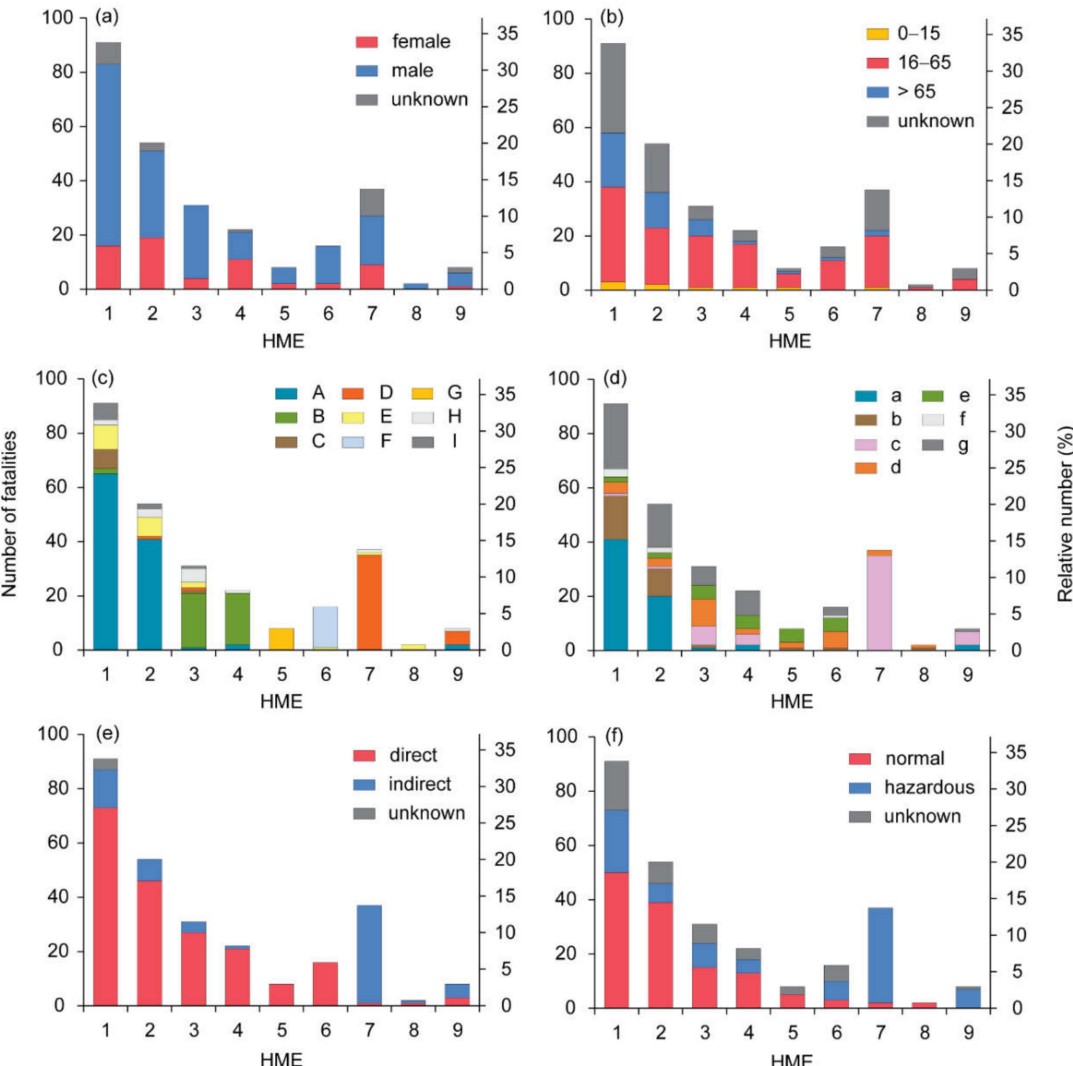

**Figure 8.** Characteristic features of fatalities expressed by their absolute and relative numbers for individual HMEs (1—flood, 2—flash flood, 3—windstorm, 4—convective storm, 5—lightning, 6—frost, 7—snow/glaze-ice, 8—heat, 9—other HMEs) in the Czech Republic during the 1981–2018 period: (**a**) gender; (**b**) age; (**c**) cause of death (A—drowning, B—tree/branch fall, C—collapse of building, D—car accident, E—health issue, F—freezing to death, G—lightning strike, H—other reason, I—unknown); (**d**) place of death (a—river/bank, b—within building, c—road/in car, d—free space in built-up area, e—landscape, f—other place, g—unknown); (**e**) type of death; (**f**) behaviour.

### 3.2.5. Type of Death (Figure 8e)

While 196 fatalities (72.9%) were classified as direct victims of HMEs, another 69 fatalities (25.7%) were evaluated as indirect; only for four fatalities was related information missing. The number of indirect fatalities compared to direct victims was higher for snow/glaze-ice calamities and slightly

prevailed also in the category of "other HMEs". For the remaining seven groups of HMEs, the situation was opposite.

### 3.2.6. Behaviour (Figure 8f)

Typical examples of hazardous behaviour leading to fatalities include those related to car accidents, often arising out of failure to adjust speed to road conditions during snow/glaze-ice calamities, attempting to ford swollen rivers and flooded low parts of roads, swimming in flooded rivers, and/or excessive consumption of alcohol followed by falling asleep outdoors during heavy frosts. Behaviour in the cases of 93 fatalities (i.e., 34.6%) was evaluated as hazardous. While another 129 fatalities (i.e., 48.0%) were classified as "normal" behaviour, sufficient information concerning behaviour was absent for 47 fatalities (i.e., 17.5%).

### 3.3. HMEs with Outstanding Numbers of Fatalities

Examples of three types of HMEs with the highest number of fatalities over the territory of the Czech Republic during the 20th century are described in detail below. As well as the two most deadly disastrous flash flood and rain-induced flood, an example of an extremely cold winter with comparable numbers of fatalities was included, together with their hydrological and/or meteorological background.

### 3.3.1. Extreme Frosts and Snowdrifts in the Winter of 1928/1929

The occurrence of frosts in the 1928/1929 winter may be elucidated in terms of the daily minimum temperatures $T_{min}$ at Prague-Klementinum (Figure 9a). The first frosty episode occurred between 14 and 26 December 1928 (21 December, −10.3 °C). The following frosty period then ran from 31 December to 7 March 1929 ($T_{min}$ was above zero only on 20 January). In January, $T_{min}$ fell as far as −19.9 °C on 12 January, and a few other episodes of deep frosts occurred in the following month, continuing for several days, with extremely low $T_{min}$ culminating on 3 February (−25.2 °C), 8 February (−19.4 °C), 11 February (−27.1 °C), 21 February (−19.9 °C), and 3 March (−21.2 °C). In other places within the Czech Lands, the corresponding temperatures were even lower than in Prague. Thus, on 11 February 1929, a $T_{min}$ of −42.2 °C was measured at Litvínovice (391 m asl) in southern Bohemia; this is the lowest temperature measured in the Czech Republic in the whole history of instrumental observations [70]. The low temperatures were also accompanied by heavy snow that created deep snowdrifts in places. Snow cover in lower positions started at the beginning of January. While in Prague-Karlov, three maxima of its depth were clearly visible, at the Luká station in eastern Moravia deep snow cover persisted at a constant level from the last third of January to the first third of March. Maxima at both stations were recorded on 28 February: 42 cm at Prague-Karlov (also on 1 March) and 50 cm at Luká.

Severe frosts with deep snow cover, strong winds, snowdrifts, and dense fog during the 1928/1929 winter led to a total of at least 48 fatalities in 45 places within Czech territory (Figure 9b): three fatalities were recorded in December, 14 in January, 25 in February, and finally six in March. Males suffered 36 fatalities, while only 10 females died (two fatalities without specification of gender). In terms of the age of fatalities, 39 were in the adult category, five were children, and three were seniors (one unknown). The main cause of death was especially heavy frost during the night, often combined with snowstorm or snowdrifts, when people lost their way; at least 15 victims were found beyond the bounds of settlements, on fields, roads, in forests, etc. In some cases, men the worse for drink fell asleep outdoors and froze to death. Not everyone was so foolish, merely poor, frozen to death in their homes when they had insufficient fuel for heating. On the evening of 19 February, a security guard at a cement-works in Olomouc-Hodolany caught three poor women stealing coal and even shot one of them [71]. A relatively high number of nine fatalities was associated with railway transport (train accident, people killed by passing train during maintenance work, crossing the rails). Five fatalities from the Krkonoše Mountains may be classified as hazardous behaviour. Several recorded fatalities

may be classified as indirect. Even criminal behaviour occurred, when two women put their new-born children outside and left them to freeze to death.

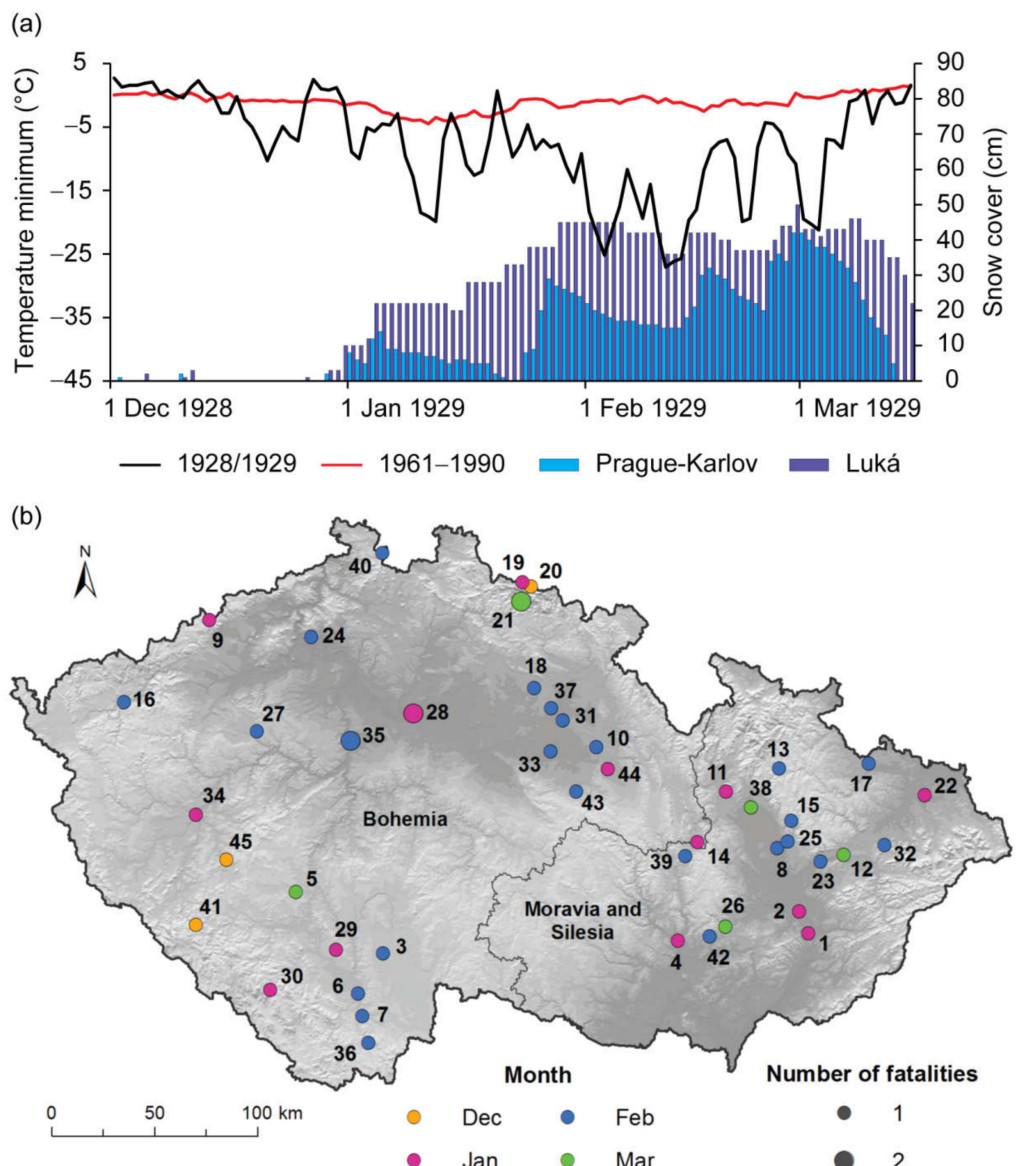

**Figure 9.** (**a**) Fluctuations in daily temperature minima $T_{min}$ at the Prague-Klementinum station ($T_{min}$ from 1 December to 15 March in the 1961–1990 reference period for comparison with $T_{min}$ in 1928/1929 appears in red) and daily depths of snow cover at the Prague-Karlov and Luká stations for 1 December 1928–15 March 1929; (**b**) places with fatalities attributable to frosts or snowdrifts in the Czech Lands for individual months: 1—Bělov, 2—Bílany (today Kroměříž), 3—Bošilec, 4—Brno, 5—Cerhonice, 6—České Budějovice, 7—Dolní Stropnice (today Římov), 8—Hodolany (today Olomouc), 9—Hora Svaté Kateřiny, 10—Horní Jelení, 11—Hrabová, 12—Hranice, 13—Jamartice (today Rýmařov), 14—Jaroměřice, 15—Jívová, 16—Karlovy Vary, 17—Kateřinky (today Opava), 18—Klenice (today Stračov), 19—Krkonoše Mts. (near Martinova bouda), 20—Krkonoše Mts. (near Špindlerova bouda), 21—Krkonoše Mts. (without specification), 22—Kunčice (today Ostrava), 23—Lipník nad Bečvou, 24—Litoměřice, 25—Lošov (today Olomouc), 26—Luleč, 27—Lužná, 28—Lysá nad Labem, 29—Malešice (today Dříteň), 30—Milešice (today Volary), 31—Nový Hradec Králové (today Hradec Králové), 32—Nový Jičín, 33—Pardubice, 34—Plzeň, 35—Prague, 36—Soběnov, 37—Svobodné Dvory (today Hradec Králové), 38—Uničov, 39—Vanovice, 40—Varnsdorf, 41—Velhartice, 42—Viničné Šumice, 43—Vrbatův Kostelec, 44—Vysoké Mýto, 45—Žďár (today Ždírec).

### 3.3.2. Flash Flood on 9 June 1970

On 9 June 1970, southeastern Moravia suffered a disastrous flash flood after a heavy cloudburst in the area. The only CHMI meteorological station in the core region, Ždánice (228 m asl), recorded 133.6 mm of precipitation, which fell between 4:40 and 7:20 p.m. Central European Time (CET), while between 3:50 and 8:00 p.m. CET, an observer recorded a thunderstorm over the station and another at a distance. Despite the fact that the stations located in the nearest proximity measured much lower totals (maximum 39.3 mm at Kyjov, 200 m asl, to the southeast, and 34.3 mm at Koryčany, 290 m asl, to the north-east), totals in the core area were probably even higher than in Ždánice (Figure 10a); for example, Cyroň and Kotrnec [72] mentioned a total of c. 195 mm for a period of 2 h. The cloudburst gave rise to a heavy flash flood, leading to disaster for the Dukla lignite mine in Šardice. As mentioned by Mika and Hurt [73], the sheer quantity of water created a large lake at the mine, with the raging river broadening to an extraordinary 102 m with the water 2 m deep. There followed a sudden burst of water, mud, and sand into the mine galleries; 32 km of corridors and shafts, together with all the workplaces and machinery, were flooded. The ground was undermined in many places, collapsed, and created many craters, some of them c. 30 m broad and deep (Figure 10b). Water flew farther out of the mine, destroying everything that stood in its way. A total of 34 miners perished in the flooded mine despite the concentrated efforts of over 200 rescuers called to the scene. The lost miners came from 13 settlements. A total of 16 victims were in the 20–29-year age range, six in the 30–39 range, and 10 in the 40–49 range (a 19-year-old youth was the youngest victim and a 53-year-old the eldest). Because the shafts were filled with transported material, extrication of the dead took a long time; by 18 June, seven miners were found, by 20 June one more, a further seven by 25 June, 18 more by 26–27 August, and the last one on 15 November. A three-year-old girl, who died at Kyjov-Boršov, had to be added to the 34 reported victims of this disastrous flash flood [74].

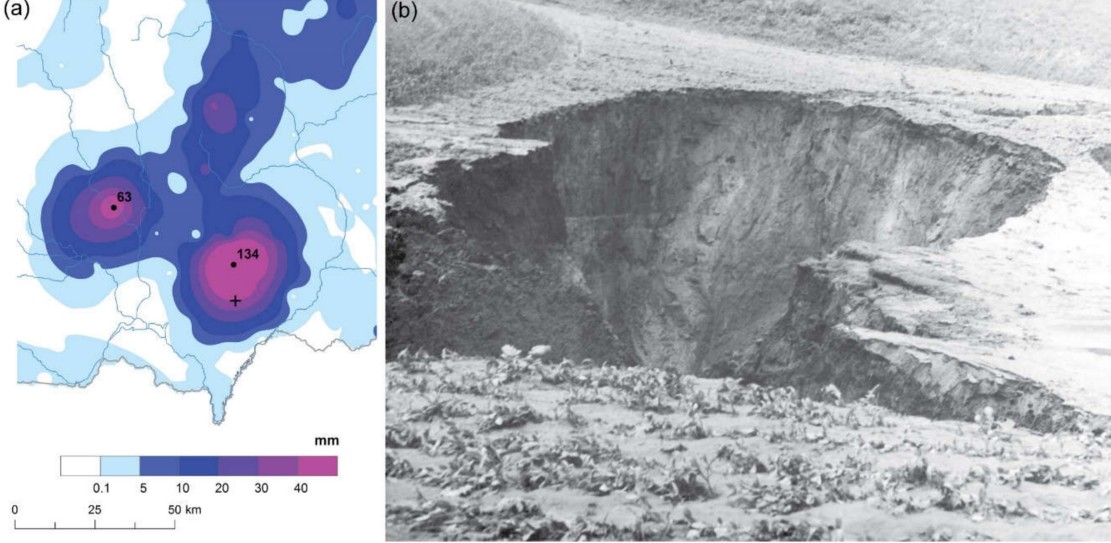

**Figure 10.** (**a**) Spatial distribution of precipitation on 9 June 1970 in the southeastern Moravia (maximum totals in two core areas: 63 mm at Brno-Kníničky (240 m asl), 134 mm at Ždánice; the + symbol indicates Šardice); (**b**) a crater created by the collapse of the upper part of the Dukla mine at Šardice [72].

### 3.3.3. Floods of July 1997

A cyclone of Mediterranean origin passing along the Vb track over central Europe (see e.g., [75,76]) brought extremely high precipitation totals to the eastern part of the Czech Republic (Moravia and Silesia) on 5–8 July 1997, particularly into the regions of the Hrubý Jeseník Mountains and the Moravskoslezské Beskydy Mountains (Figure 11a). For example, on 6 July, four stations measured more than 200 mm (Lysá hora Mt., altitude 1324 m asl, had a total of 233.8 mm) and nine stations more

than 150 mm. This resulted in floods on many rivers, and a particularly critical situation developed on 7–8 July. Based on measurements taken by CHMI hydrological stations, on 7 July, peak discharges recalculated to N-year return period were $> Q_{100}$ for 16 stations, $Q_{100}$ at one station, and $> Q_{50}$, $> Q_{20}$, and $Q_{20}$ at two stations each. The corresponding numbers for 8 July were $> Q_{100}$ for seven stations, $Q_{100}$ for two stations, $> Q_{50}$ for one station, $Q_{50}$ and $> Q_{20}$ for two stations each, and $Q_{20}$ for three stations. This means that peak discharges of $\geq Q_{20}$ were recorded at 40 hydrological stations in Moravia and Silesia during these two days. Over the following days, sites with discharges of $> Q_{100}$ moved downstream along the River Morava [39,64,77].

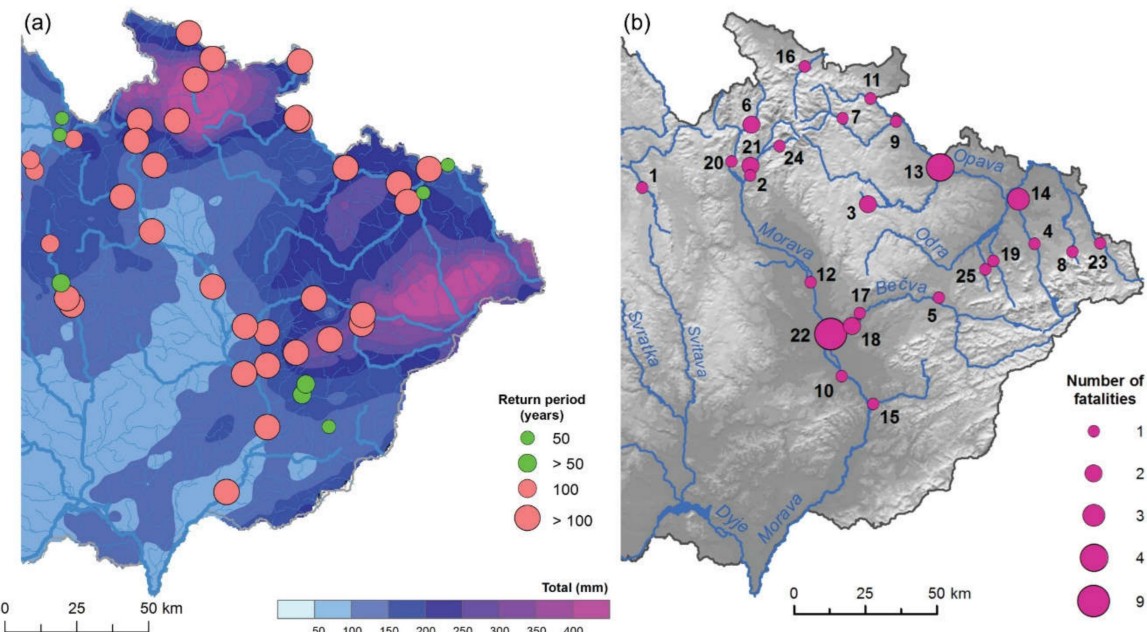

**Figure 11.** (**a**) Spatial distribution of precipitation totals on 5–8 July 1997 in the eastern Czech Republic and N-year return period of peak discharges $Q_N \geq 50$ years for Czech Hydrometeorological Institute hydrological stations; (**b**) location of places with documented numbers of fatalities during the 1997 flood: 1—Česká Třebová, 2—Dolní Studénky, 3—Dvorce, 4—Frýdek-Místek, 5—Hustopeče nad Bečvou, 6—Jindřichov, 7—Karlovice, 8—Komorní Lhotka, 9—Krnov, 10—Kroměříž, 11—Město Albrechtice, 12—Olomouc, 13—Opava, 14—Ostrava, 15—Otrokovice, 16—Písečná, 17—Prosenice, 18—Přerov, 19—Příbor, 20—Ruda nad Moravou, 21—Šumperk (district), 22—Troubky, 23—Třinec, 24—Vernířovice, 25—Závišice.

Different sources mentioned a range of numbers of fatalities. Although 60 fatalities may be accepted as the upper limit [78], newspaper information enabled detailed accounts of only 42 fatalities, in 24 specific locations and one district (Figure 11b). The village of Troubky became a symbol of this particularly disastrous event. Eight people died there during the night of 7/8 July; the majority of them drowned when their houses collapsed. The ninth victim, a 76-year old woman, died of hypothermia a few days later despite hospital care; she never recovered from spending too many hours in cold water (see Figure 3a). With the exception of one fatality in Česká Třebová (a man jumped into the flooded river intending to swim, but drowned [79]), all other fatalities occurred in Moravia and Silesia. Males (26) accounted for most of the losses among the 42 fatalities, while eight females died; there were further (eight) "unknown" fatalities. In terms of age structure, 18 fatalities were categorised as adult, 12 as seniors, and a further 12 remained unclassified. Drowning was identified as the main cause of death for 26 victims, and four further people were killed by collapsing buildings. While for six victims the cause of death is unknown, heart failures were responsible for four fatalities. Some fatalities resulted from hazardous behaviour, similar to that reported above for Česká Třebová. The fast train

from Vienna to Warsaw was derailed by waterlogged track at Suchdol nad Odrou on 7 July; fortunately, no one died but 64 people were injured, 10 of them seriously [80].

## 4. Discussion

### 4.1. Data Uncertainty

As is generally known from work with documentary evidence in historical climatology [49,50] and in historical hydrology [51,52], results suffer from limitations on the extent and availability of existing documentary sources, as well as from a lack of research capacity for the extremely time-consuming collection of data. These may be reflected in spatial and temporal inhomogeneity of the data gathered after critical evaluation; all this holds true for information concerning fatalities arising out of HMEs. As follows from Figure 5, the numbers of fatalities, as well as related HMEs, are clearly underestimated before 1997 and after 2011 (despite a meticulous search of internet sources for 2002–2016 [81]), something that holds especially true for years with no documented fatality. However, even in the most complete period (1997–2010), documented fatalities are undervalued, as is evident from the figure of only 42 fatalities for the July 1997 flood documented in this study compared to the 60 fatalities reported by Punčochář [78].

Critical evaluation constitutes an essential aspect of the use of fatality data from documentary sources. For the purposes of this paper, particular attention and preference were given to information reporting name, age, and the cause of death in great detail for each fatality. We consider the use of only summary information reporting the total number of fatalities during the event somewhat problematical, although it is typical of many papers and studies addressing disastrous HMEs, in which details of material damage tend to take precedence. When fatalities are reported by different sources without the necessary details, the result may be an underestimation of real numbers on the one hand, or even in the exaggeration of fatalities on the other. In some cases, even already reported fatalities were declared untrue later.

Despite a critical approach to information from documentary sources, there remain cases in which it is difficult to decide what is really correct. Particularly unclear are situations in which people seriously injured in the course of HMEs were transported to hospital (e.g., after a lightning strike or falling of trees/branches during windstorms); sources seldom provide any additional information about when and if they recovered. This implies that newspaper sources are sometimes insufficient for such study, and medical statistics and databases will have to be explored in future research.

Furthermore, two cases of fatal heart attack attributed to extreme heat were reported in newspapers and were included among the fatalities recorded in the Czech Republic for the 1981–2018 period. However, this type of fatality requires the use of mortality data prepared by the Czech Statistical Office and the Institute of Health Information and Statistics of the Czech Republic, which was the approach taken by Urban et al. [82] in a study analysing mortality during the heat waves of 2015 in comparison with other outstanding heat events in 1994, 2003, 2006, 2010, and 2013 (see also [83–85]).

### 4.2. Temporal Changes in Factors Affecting Fatality

The collection of information concerning fatalities in Czech territory during the 20th and 21st centuries shows the effects of certain temporally changeable factors confined not only to climate variability and HME changes, but also to political, socio-economic, and societal changes, as the following examples demonstrate.

#### 4.2.1. Lightning Fatalities

The low number of eight documented fatalities due to lightning strike in the 1981–2018 period is not immediately comparable to the numbers of fatalities extracted from the newspaper *Lidové noviny* for the Czech territory between 1 April and 30 September in the following years: 1903—nine fatalities, 1904—21, 1905—20, 1906—25, 1907—14; mean 1903–1907—17.8 fatalities per year. This disparity

reflects changes in work practices, with a far higher number of people working in agriculture and forestry in the past than at present, as well as the fact that people were more frequently outdoors and exposed in the open landscape. Lightning used to kill more people indoors before the advent of the almost universal use of lightning conductors. Vast improvements in medical services and communications led to a marked decrease in lightning fatalities; more or less immediate emergency help is available, while telephones of all types provide links to rapid transport to a broader network of hospitals. Public awareness of how to behave during a thunderstorm also plays a positive role. A considerable decline in the number of lightning fatalities was also reported in Switzerland when the 1946–1975 and 1976–2015 periods were compared [31].

### 4.2.2. Transport Accidents Related to the Weather

This category usually includes indirect fatalities attributable to extreme weather conditions, related among other factors to the ever-burgeoning use of road vehicles both private and commercial in preference to public transport (e.g., train). However, six men died on 4 May 1911 when a heavy downpour led to a layer of soil and stones sliding onto the railway tracks near Ústí nad Orlicí, tipping a locomotive into a slope [86]. Particularly critical in relation to the density of modern road transport are snow or glaze-ice conditions, when cars driven inappropriately skid off the road, or into collisions, resulting in possible fatalities. Although 35 such fatalities were recognised in 1981–2018, the real number would be higher, since snow/glaze-ice is not always mentioned as the primary reason for the skid. Various traffic accident databases have much potential. Some are maintained by the Czech police or were newly established on various web servers as public services. This category may be expanded to include accidents arising out of, for example, dense fog, falling trees or parts of them onto roads during windstorms, heavy rain, etc. The majority of cases may be evaluated as hazardous behaviour on the part of many drivers, who fail to make sufficient allowance for extreme weather conditions.

### 4.2.3. Social Bias in Fatalities

Information concerning fatalities attributable to HMEs may be set aside from the category of societal interest in the light of significant socio-historical events. The two world wars in 1914–1918 and particularly in 1939–1945, for example, occupied the bulk of human attention, and HME fatalities fell into the background in the light of the consequences of war. A similar distraction of public attention from HME fatalities also appeared in the past political unit known as Czechoslovakia during the totalitarian regime that held power in 1948–1989, when the propagation of communist ideology in the media took absolute priority (e.g., publication of voluminous political speeches). The consequences were already pointed out by Brázdil et al. [87], who reported on the general disappearance of reports of meteorological extremes in Czech newspapers after the communist takeover in 1948, attributing it to the communist cliché, "we shall command the wind and the rain"; this continued until the late 1950s. Even afterwards, similar information was somewhat suppressed, or mentioned only as an aside, in the form of second-rate information with no corresponding details.

### 4.3. The Broader Context

There exists no unified system for collating evidence of the impacts of disastrous natural phenomena in the Czech Republic. Some kind of evaluation appears only in specialised reports after disastrous events such as floods or flash floods with great material damage and loss of human lives. This means that certain individual efforts dedicated to the creation of databases of such events appear very constructive, also with respect to the priorities and targets reported in the SFDRR (see [5]).

That the highest levels of attention are devoted to floods is hardly surprising. Annual numbers of flood fatalities may be compared with those reported by Punčochář [78], and slightly detailed information, including settlement, gender, age, reason, and place of fatality, for certain floods by Brázdová [48], as shown in Table 2. The greatest difference with this paper appears for 1997, whereby 42 fatalities were documented in our study compared to 60 fatalities reported by Punčochář [78], or to

only 14 fatalities detailed by Brázdová [48]. Greater or lesser differences in the number of fatalities also appear in other years. While Punčochář [78] reported a total of 135 fatalities for the 1997–2013 period, our study specifically documented 129 fatalities.

**Table 2.** Comparison of numbers of flood fatalities in the Czech Republic reported in different sources for the 1997–2013 period: A—Brázdová [48], B—Punčochář [78], C—this study.

| Source | Year | | | | | | | | | | | | | | | | |
|--------|------|------|------|------|------|------|------|------|------|------|------|------|------|------|------|------|------|
|        | 1997 | 1998 | 1999 | 2000 | 2001 | 2002 | 2003 | 2004 | 2005 | 2006 | 2007 | 2008 | 2009 | 2010 | 2011 | 2012 | 2013 |
| A | 14 | - | - | 2 | - | 17 | - | - | - | 11 | - | - | 18 | 8 | - | - | - |
| B | 60 | 10 | 0 | 2 | 0 | 16 | 0 | 0 | 0 | 9 | 0 | 0 | 15 | 8 | 0 | 0 | 15 |
| C | 42 | 7 | 0 | 3 | 1 | 21 | 2 | 0 | 2 | 7 | 0 | 0 | 15 | 11 | 0 | 1 | 17 |

Daňhelka [88] reported, in the context of an SFDRR meeting, the creation of a CHMI database of historical disaster phenomena and their impacts since 1993 (83 events, 235 fatalities, divided into 126 males, 47 females, and 62 unknown). For the 2005–2015 period, he mentioned 93 fatalities: 42—floods, 21—flash floods, 16—frost and snow, nine—windstorms, six—lightning, six—avalanches (note that the sum of these is 100), and two fatalities due to landslides in 2013. For the same period, the current contribution recorded 111 fatalities: 27 each—floods and flash floods, 28—frost and snow/glaze-ice calamities, 19—windstorms and convective storms, six—lightning and avalanche, and four—other HMEs. Notable differences appear (with the exception of flash floods and lightning fatalities) in all categories.

The flood fatalities in the Czech Republic collected for this study were used by Petrucci et al. [28] for the 1980–2018 period in the context of eight other European regions (particularly the Mediterranean) to analyse their different features in detail, based on the EUFF database. While Czech flood fatalities, together with Portugal, the Balearic Islands, and Israel, showed stable linear trends, Greece, Italy, and southern France indicated increasing trends, while Turkey and Catalonia (Spain) exhibited decreasing trends. Among 2466 detected flood fatalities, the male category, at ages of 30–49 years, with the majority of deaths outdoors, prevailed. Deaths in cars carried away by water/mud appeared most frequently. Drowning was a primary cause of death, followed by heart failures. These findings tally closely with the results of this Czech study.

However, a comparison of Czech fatality results with Switzerland appears more appropriate. It is another central European country, despite the fact that Badoux et al. [31] analysed a substantially longer period: 1946–2014. Even so, compared with a mean of 7.1 Czech fatalities per year, this doubles in Switzerland (14.7). The existence of the Alps places fatalities due to snow avalanches in the first place (37%), followed by lightning (16%), floods (12%), windstorms (10%), rockfall (8%), landslides (7%), and other processes (9%). While disasters of a geomorphological character were not taken into account in the Czech fatalities chronology, the relative proportions of flood and windstorm fatalities were significantly higher (54% and 20%, respectively), while, for lightning, the number was only 3%. This shows that, even between relatively close countries, but with different natural patterns prevailing, the differences in the numbers and structure of fatalities can be quite substantial. After further supplementation of the preliminary Czech fatality database, there will be ample opportunity for future comparison with the comprehensive Austrian torrential event catalogue (see [35]) to extend our knowledge of HME-related fatalities on a broader central European scale.

Two of three events with outstanding numbers of fatalities in the Czech Republic discussed in Section 3.3 also impacted upon other central European countries. In particular, the July 1997 flood also affected Poland, Slovakia, and Germany (with 54 fatalities reported for Poland [89]). The severe winter of 1928/1929 [90] occurred on a broad European scale, but detailed data on fatalities at the European or individual country levels did not appear.

### 5. Conclusions

This paper addressed the potential of documentary evidence in the description of fatalities attributable HMEs, demonstrated through the example of the 1981–2018 period and three outstanding events in the 20th century in the Czech Republic. The main results may be characterised as follows:

(i)   The uncertainty in documentary data used for the creation of the Czech fatality database provided only underestimated numbers of fatalities, as well as of HMEs involving fatalities. Further supplementation of this database is required, particularly with data from before 1997 and after 2011.

(ii)  Floods and flash floods were the predominant types of HME giving rise to fatalities in the Czech Republic, followed by windstorms with convective storms and snow/glaze-ice calamities. The other HMEs considered did not achieve comparable importance. The majority of fatalities concentrated around the summer months, followed by winter.

(iii) Males and adults clearly prevailed over all other categories of fatalities classified herein. The primary cause of fatalities was drowning, associated in terms of place of death with rivers and/or their banks. Around one-quarter of fatalities may be considered indirect, and one-third were consequences of hazardous behaviour.

(iv)  Joining fatality trends (caused by HMEs) with climate variability/change requires the most complete database of fatalities. Temporally changeable factors influencing fatalities also have to be taken into account. The preliminary Czech fatality database, for the 1981–2018 period, does not yet permit the formulation of any such consideration with respect to observed climate change.

(v)   Documentary evidence constitutes an important source for the creation of databases of fatalities attributable to HMEs for the most recent period, as well as for the past. Newspaper reports are the most important source. The use of documentary data opens a new field of application for historical–climatological and historical–hydrological data worldwide.

(vi)  The experience and results gathered in this contribution will be further used for extensive research, in order to create a 1901–2018 chronology of fatalities attributable to HMEs over the territory of the Czech Republic. As well as traditional documentary evidence, new sources and partial databases kept by a range of institutions (e.g., the health service, the police, the fire-fighting service) will have to be used.

This study is of notable originality in terms of the territory of the Czech Republic. Furthermore, it addresses fatal HMEs on a broad scale, discussing the weaknesses and strengths of documentary data for this type of study and extending knowledge of HME-induced fatalities to other parts of Europe. All of these factors should be considered novel aspects.

**Author Contributions:** Conceptualization, R.B.; methodology, R.B., K.C., and J.Ř.; validation, K.C., J.Ř., P.Z., and L.D.; formal analysis, R.B. and K.C.; investigation, P.Z., L.D., L.Ř., and P.D.; data curation, J.Ř., P.Z., L.D., L.Ř., and P.D.; writing—review and editing, R.B.

**Funding:** This research was funded by the Ministry of Education, Youth, and Sports of the Czech Republic for the SustES—Adaptation strategies for sustainable ecosystem services and food security under adverse environmental conditions project, ref. CZ.02.1.01/0.0/0.0/16_019/0000797.

**Acknowledgments:** Jan Řehoř was supported by Masaryk University within the MUNI/A/1576/2018 "Complex research of the geographical environment of the planet Earth" project. We acknowledge the help of P. Štěpánková (Brno) and P. Raška (Ústí nad Labem) in providing us with some special information related to HMEs. Tony Long (Svinošice) helped work on the English. We would also like to thank the three anonymous reviewers for their comments that helped significantly to improve the manuscript.

**Conflicts of Interest:** The authors declare no conflicts of interest.

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
