# Peer review of "Potential of Documentary Evidence to Study Fatalities of Hydrological and Meteorological Events in the Czech Republic"

_water, doi:10.3390/w11102014_

Round 1

Reviewer 1 Report

The study performs a comprehensive analysis of the deaths related to meteorological and hydrological events in the Czech Republic by making use of multiple-source documentary evidence. It is an excellent demonstration of how these data contribute to the understanding of the spatiotemporal pattern of fatalities associated with such hazards.

The manuscript is, in general, well written by the authors with a clear structure.  The objective of the study is clearly presented in Introduction and data used in the study are sufficient and relatively properly analyzed. Nevertheless, I would like to raise some comments that required a point-by-point response by the authors in the re-submission.

Major comments:

The current title needs to be shorter and more accurate in light of the objective of the study.

The current abstract does not reflect the contribution that the study makes. For example, the spatiotemporal variability of the deaths from the disasters should be briefly described. The conclusions of the study need be highlighted at the end.

I appreciate the effort that authors made to collect data from a wide range of sources. As a reviewer and reader, I would strongly recommend tabulating these data to enhance the readability. Information such as disaster type, the number of deaths, and data source can be cited in the table. This would help to gain an improved sense of what event caused how many deaths from which source(s) (where).

Technical terms such the extreme events (L231-L248) should be defined in somewhere of the manuscript in case of misunderstanding to readers as these terms are not uniformly defined in different contexts.

While I agree with you that it is critically important to check the quality of the information sources, my main concern lies in the section of Method. The method how authors evaluated the reliability is unclear to me, and this would not allow the study to be re-produced. The relative importance of these sources and how the cross-validation was conducted must be detailed in the section of Methods. Particularly, authors must make it crystal clear the resolution when there are conflicting descriptions on the death toll from the same event.

I notice that authors were aware of the social bias in fatalities but there might be cases of deaths in remote villages or in places that were not covered by the press—this would underestimate the negative impact of the HMEs. How did you address this issue?

The background of the Czech Republic should be placed in Section 2 or as an independent section of Study Area. The societal and climatic background is absolutely not a part of Results. This is just an introduction to the context and provides a basis for the study. Do remember to include some key information that would be key for analysis and discussion later in the manuscript.

It is noted that this manuscript is full of descriptive text regarding the examples of death from the HMEs. This is useful to demonstrate the consequences of the extreme events while too much is quite distracting. As such, I recommend reducing these parts to a minimal level without losing the information that authors intended to deliver.

The value of the vast amount of data collected by authors is not well-used. An in-depth analysis would be necessary to reveal the spatiotemporal variability of these HMEs related fatalities. To this end, (more) quantitative analysis and spatial analysis are recommended.

I doubt the roles of the three examples that play in Results (Section 3.3).

Overall, the novelty of the study needs be highlighted in Introduction and Discuss.

Minor comments

Incorrect abbreviation: (a) hydrological and meteorological events (HMEs) or (2) meteorological and hydrological events (MHs).

Comments on Figures: (a) coordinates, north arrow, regional borders, and major rivers/water bodies should be shown in Figure 1, 7, and 9; important river/water bodies should be labeled as they might be closely associated with some of the HMEs. (b) no relative numbers are given in Figure 4. (c) in Figure 9, “Tmin in the 1961-1990 reference period appears in red” is not understandable to me, as the X-axis only shows the days ranging from Dec 1928 to Mar 1929.

Please double-check the reference style of the journal’s reference styles.

Lastly, despite the English of the paper being quite good, some typo/grammatical errors do exist in this manuscript. For example, L226, ‘a’ not ‘an’; L541 dash should be deleted; L578 ‘number’. As such, authors might want to check these small language issues throughout.

Reviewer 2 Report

The manuscript is readable, quite lengthy due to some quotations and three HME cases that are presented into too much detail. I am wondering whether these three cases with details (such as personal names or detailed hours) bring substantially to the message of the paper. Especially, also, as two of the events are prior to the main period under investigation (1981-2018).What is the importance of bringing forward these three cases? How they relate to the modern database and climate change? How they relate to the obligations of countries to implement the Sendai Framework 2015-2030? I am of the opinion that these three cases should be omitted from the manuscript, since they are described into much too detailed way.

The paper is on one hand about history and the importance of numerous classical information sources, and on the other hand about a "modern" database (1981-2018) with its pluses and minuses.

The review in the Introduction is interesting, I may suggest to read and add some further literature into the manuscript, such as papers:

Haque et al (2017) Fatal landslides in Europe. Landslides 13(6), 1545-1554

https://link.springer.com/article/10.1007/s10346-016-0689-3

Froude MJ, Petley DN (2018) Global fatal landslide occurence from 2004 to 2016

Heiser, M., Hübl, J. & Scheidl, C. (2019) Completeness analyses of the Austrian torrential event catalog. Landslides, https://doi.org/10.1007/s10346-019-01218-3

https://link.springer.com/article/10.1007/s10346-019-01218-3

Špitalar M et al (2014) Analysis of flash flood parameters and human impacts in the US from 2006 to 2012. Journal of Hydrology, Volume 519, Part A, 27 November 2014, Pages 863-870.

https://www.sciencedirect.com/science/article/pii/S0022169414005216

and you can use these references also in section 4. Discussion, where there is very few correlations with the described database in the Czech Republic with other Central European databases (apart from the total number of fatalities in Switzerland). This section 4. Discussion must be improved by referencing also for example the Austrian database, and discuss how your database lacks or lacks not of the completeness, and what are the reasons for that and what are the solutions to have a more complete database. What about the old Austro-Hungarian monarchy and their torrent control service that was established in all countries at that time?

Specific comments are:

line 39 - HMEs cause? great material damage...

line 122 - ... in this paper (given in a alphabetical order):

line 231 - use rainfall-induced floods instead rainy floods

line 232 - use melting instead of thawing

line 242 - ... by a direct strike.

line 246 - ..., usually during heat-waves.

line 251 - replace ....in a particular country... by...in the Czech Republic... or rephrase.

line 263 - can you give also interval of mean annual temperatures, mean annual precipitations etc, e.g. mean annual temperatures in the Czech Republic are from xx to yy degrees C, on average 8,1 degrees C, .....

line 279 - this section 3.2 Spatiotemporal variability of fatalities has simple figures (4 to 8) and lengthy description what can be seen on these figures. The text should bring some more flavour, or please, make the text shorter, as e.g. the lines 295 to 302 can be written as: The fatalities are unevenly distributed in the studied period (1981-2018), with 10 years without fatalities, and another 10 years with only one or two fatalities.

line 494 - The corresponding numbers....not figures

line 577 - The low number of 8... not figure

line 660 - ... for lightning the number was only 3 %.

line 768 - should be Zêzere

Reviewer 3 Report

In the discussion, one can refer to the effects of HME in other countries of the region, e.g. the flood in Poland in July 1997 (so-called flood of the millennium) - 56 fatalities

Round 2

Reviewer 1 Report

I am satisfied with the revisions that the authors made to address my comments.

Reviewer 2 Report

Dear Authors,

thank you for a revised manuscript where you considered the proposed comments and suggestions to improve the original manuscript.

From my point of view, the revised manuscript is now acceptable for publication.

Best regards